# Application and Development Progress of Cr-Based Surface Coating in Nuclear Fuel Elements: II. Current Status and Shortcomings of Performance Studies

**Huan Chen, Xiaoming Wang and Ruiqian Zhang** *

Science and Technology on Reactor Fuel and Materials Laboratory, Nuclear Power Institute of China, Chengdu 610200, China; npicchh@163.com (H.C.); npicwxm@163.com (X.W.)

*  Correspondence: zhang_ruiqian@126.com or fanyingduiranliaojicailiaozhongdianshiA@npic.ac.cn

**Abstract:** In continuation from the earlier Part I, this paper presents the status on the investigations of Cr-coated zirconium alloys under both normal operating and hypothetical accident conditions. This paper is aimed to provide sufficient knowledge to understand the service performance and research focus of accident-tolerant fuel (ATF) coatings.

**Keywords:** zirconium alloy; accident tolerant fuel; ATF cladding material; Cr-based coating; service performance

---

## 1. Introduction

From the introduction in Part I, it is clearly understood that the accident-tolerant fuel (ATF) concept is designed to significantly alleviate or avoid heat and hydrogen release generated due to severe zirconium (Zr)–water reaction owing to the loss-of-coolant accident (LOCA) in the reactors, and to delay the embrittlement failure of the cladding during the quenching process, thereby finally increasing the safety allowance of nuclear fuel elements. It is a simple and effective method to obtain a protective coating on the surface of Zr alloy with resistance to high-temperature oxidation. Notably, chromium (Cr)-based coating demonstrated impressive application prospects and thus has attracted the attention of researchers worldwide [1]. However, the engineering application of coated cladding requires not only improving the performance under accident conditions, but also at least maintaining (if improvement is impossible) the performance under normal operating conditions. Therefore, it is somewhat necessary to carry out performance evaluation and investigations based on these two aspects.

The foremost concerns the performance of the Zr alloy with Cr-based coating including its mechanics, corrosion, fretting wear, and irradiation damage behavior under normal operating conditions and high-temperature oxidation; embrittlement upon quenching; and LOCA-transient ballooning/bursting behavior under accident conditions. The objective of Part II of this review is to introduce the current status of the study on the performance of Cr-coated Zr alloys in order to understand the above behaviors related to Cr-based coating. Moreover, the focus is directed towards coatings' protective effects on Zr alloys, their interactions, the coating consumption and degradation during protectivity evolution of the coating, and the possible mechanism behind these phenomena. Finally, the shortcomings of the current studies are analyzed.

## 2. Performance under Normal Operating Conditions

### 2.1. Mechanical Behavior

The impact of the coating itself or the preparation process on the mechanical properties of the Zr alloys matrix is a primary concern in the service of the Zr alloy coatings. In Part I of the review, an analysis on the characteristics of Cr-based coating by different methods of preparation manifested that the microstructure of Zr alloy matrix was not significantly influenced by the coating process (especially in the case of physical vapor deposition (PVD)) when the temperature of preparation was lower than the recrystallization temperature of Zr alloy [1]. Under normal operating conditions, the thin external Cr coating has little impact on the overall mechanical properties of the cladding, because the mechanical strength of pure Cr did not markedly differ from that of industrial Zr alloys (from room temperature to the cladding temperature of about 350 °C) [2]. However, the Cr-coated Zr alloy cladding prepared by 3D laser melt-coating (3D-LMC) increased the hoop strength, which may be due to on the one hand the relatively large thickness of the coating (~80 μm), and on the other hand the rapid solidification microstructure (such as martensite) formed by this process [3]. Besides, Kim et al. [4] simulated the hoop compression mechanical properties of the three types of ring specimens with a length of 10 mm and out diameter of 9.5 mm in finite element analysis (FEA), including uncoated Zr-4 (wall thickness: 570 μm) and Cr (80 μm) + Zr-4 (570 μm) as well as Cr (80 μm) + Zr-4 (490 μm). The results revealed that even with the thinning of cladding wall, the ductility of Cr-coated Zr-4 cladding could still be maintained or enhanced as the high elastic modulus of Cr coatings allows it to absorb the extra stress.

During normal operation, axial growth and hoop creep deformation always occur to the Zr alloy cladding, of which, the former is due to the irradiation-induced anisotropic growth of the hexagonal close-packed (hcp) structure of Zr alloys, while the latter results from the pressure difference, caused by the external high-pressure cooling water and the internal fission gas release, and also the pellet–cladding interaction (PCI) [5]. Therefore, special attention is required to the possible spalling and cracking of the coating along with the deformation of the matrix, as they may destroy the structural integrity and weaken the chemical protection of the coating. Coating spalling is mostly affected by the adhesion quality, and the intrinsic adhesion quality of the coating is determined by the process of preparation. From the results of the current study, spalling of the Cr coating from the surface of the Zr alloy substrate has not been reported. For example, Kim et al. [3] studied the adhesive strength between Zr matrix and Cr coatings through ring tensile and compression test, and it was found that the interfacial adhesion remains the same without any occurrence of spalling or buckling-driven delamination, even under high compression or tensile strains (up to 6%). Brachet et al. [2] and Shah et al. [6] stretched the Cr-coated Zr alloy sheet and cladding tube axially until the matrix broke and failed, and no indication of coating spalling was observed. This shows that Cr coatings obtained by the mainstream preparation methods (as described in Part I of this review [1]) have overall adhesion quality, which can be attributed to the formation of the Zr-Cr diffusion layer at the interface. For example, Ribis et al. [7] revealed the crystalline coherence in the interfacial diffusion region of PVD Cr-coated Zr alloys through atomic-scale characterization with high resolution. However, for preparation by following methods such as PVD, CS, and 3D-LMC, cracking in Cr coatings under mechanical load was observed [2,3,6]. To confirm the ultimate strain of cracking in the coating, Kim et al. [3] conducted 2%, 4%, and 6% hoop tensile strain tests on Cr-coated cladding samples prepared by 3D-LMC. The results showed that the Cr-coated cladding could withstand 4% strain without cracking, but when the strain reached 6%, a series of radial cracks occurred in the Cr coating. Still, it meets the requirements of resistance to 1% hoop strain of the specifications of fuel claddings [8].

Wagih et al. [9] predicted the mechanical behavior of Cr-coated Zr-4 cladding under PCI and transient conditions during normal operation through the multi-physics fuel performance tool, Bison. The study indicated that under steady-state normal operating conditions, due to higher hardness and lower thermal creep of Cr coating, the inward creep of Cr-coated cladding is smaller, which delays the

mechanical contact between the fuel and the cladding. Due to thermal mismatch with the Zr-4 matrix, it yields Cr coating upon initial heating from room temperature to 1.3% plastic strain by the end of life (EOL). However, some data showed that the elongation of Cr could reach 40% before fracture, which occurs at 300 °C. Therefore, we believe that the Cr coating will not break at the EOL. During the transient period of power ramp, the stress of the Zr-4 matrices of the Cr-coated cladding was reduced by nearly half compared to the uncoated (as a reference), which is mainly due to the lower contact pressure of pellet cladding. In all cases, the coating always produced before the cladding. Generally, Cr-coated cladding shows excellent performance, but the plastic strain may adversely affect the performance of the Cr coating.

Lee et al. [10] predicted the stress distribution and evolution with burn up of the coated cladding in the reactor through numerical analysis while taking the impact of pressure difference, thermal expansion, irradiation growth, and creep into consideration. The results exhibited that, in the coating, the axial irradiation growth and creep deformation led to tensile stress and compression stress, respectively. Depending on the coefficient of thermal expansion (CTE), Zr matrices and coatings can be compressed or stretched. As the CTE ($\mu m \cdot m^{-1}\ {}^{\circ}C^{-1}$) of Cr (6.2) is similar to Zr alloys (6.0), the outer region (lower temperature) with less thermal strain is subjected to self-equilibrium tensile stress. Increasing the coating thickness can effectively reduce the stress level of the coating itself, but it can also increase the stress level of the Zr matrix. When the coating is too thick, it may cause Zr alloys to deviate from ideal behaviors, which is considered to be a potential safety issue. Generally, excessive stress in the coating leads to increasing the stress in the coated-Zr alloy system, which is chiefly caused by the axial growth and creep mismatch between the coating and Zr matrix. Moreover, the stress level in the coating increases sharply from the beginning of life (BOL) to the middle of life (MOL). From the MOL to EOL, the stress remains unaffected. This is due to that from BOL to MOL, irradiation growth and creep deformation slowly increase, which significantly deteriorate the structural integrity of the coating. While from MOL to EOL, creep deformation compensates for irradiation growth. Except for BOL, the stress level of the coated Zr alloys is several times higher than that of uncoated Zr alloys.

Ševeček et al. [11] experimentally simulated the acceleration of inward creep and outward displacement of the cladding after the PCI of the cold spray (CS) Cr-coated Zr-4 cladding tube samples in the presence of high-pressure dynamic water. The results validated the reliability of the theoretical prediction mentioned above that due to the mismatch of the underlying physical properties of the layers, higher stress could be generated, and the predicted plastic strain could lead to cracking in the coating. They also studied the fatigue behavior of CS Cr-coated Zr-4 sheet samples through four-point bending tests under normal pressure dynamic water conditions [12]. The bottom surface of the sample was stretched, whereas the top surface was compressed. It was found that due to high-temperature creep, the specimen was bent. The top surface of the coated sample was free of defects with good adhesion of the coating, but there were many cracks on the bottom surface. The specimen failed at the pin position. The initiation time for the cracks in the coated sample was less than in the uncoated reference sample, as shown in Figure 1. It is believed that the initiation of early cracks may be linked to stress concentration owing to nonuniformity in the CS coating, which might lead to the reduction in the fatigue life of the coated parts.

Given the possible cracking behavior of Cr coatings under normal operating conditions, Hong et al. [13] calculated the internal stress and channel crack propagation conditions of Cr coatings through establishing the finite element model of fuel pellet cladding system and using linear elastic fracture mechanics (LEFM). It was found that the stress in the Cr coatings is mainly determined by its interaction with the Zr matrices, as the Cr coatings are much thinner than the Zr alloy matrices. When the cladding expands outward through PCI, a high tensile hoop stress is generated in the coatings. The maximum hoop stress increases with an increase in the linear heating rate (LHR). When the creep relaxation of Cr coatings is dominated by the dislocation glide mechanism (at high LHR), the maximum hoop stress becomes independent of LHR. Besides, under any given power conditions, the stress in the coatings is independent of the coating thickness, which causes the steady-state energy release rate

of the channel crack to linearly increase with an increase in the coating thickness. When the energy release rate is higher than the fracture toughness of Cr, the cracking of Cr coatings occurs. The resulting critical coating thickness decreases with an increase in the LHR, which reaches a plateau at a high LHR, as shown in Figure 2. It has been pointed out that the coating thickness should be less than the critical value in the design of Cr-coated Zr alloy cladding.

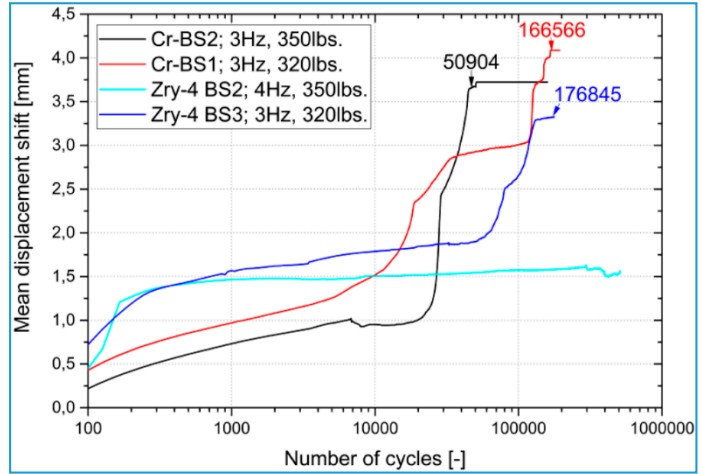

**Figure 1.** Comparison of fatigue behavior of Cr CS-coated Zr-4 sheet and uncoated samples. Reproduced from the work in [12] with copyright permission from the American Nuclear Society (ANS) and the author.

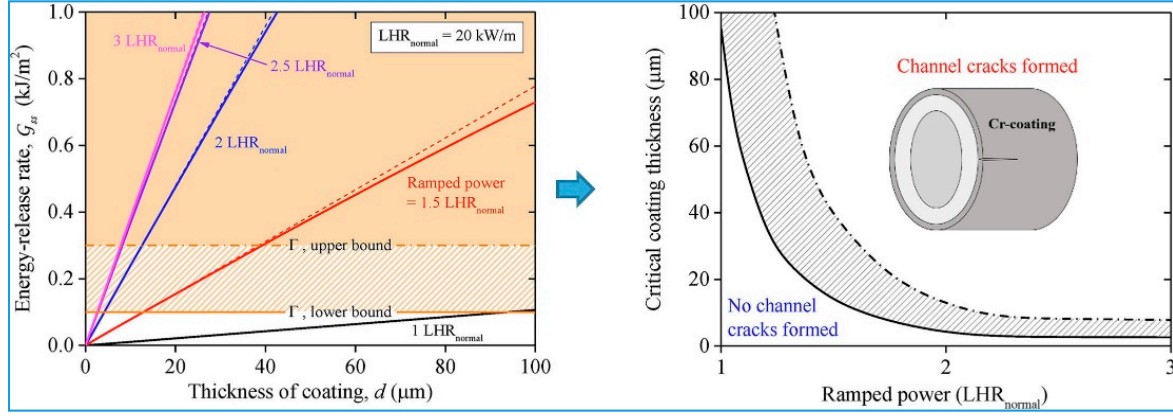

**Figure 2.** Functional relationship between critical coating thickness and ramped power. Reproduced from the work in [13] with copyright permission from Elsevier.

### 2.2. Corrosion Behavior

The waterside corrosion of Zr alloy cladding and the accompanying hydrogen absorption behavior affect the reliability of the fuel cladding and limit further improvement in the consumption of reactor fuel, which is one of the vital concerns related to performance [5]. Existing studies have shown that Cr coatings enhance the corrosion properties of Zr alloys. However, the improvement is subject to the quality of the coating. Brachet et al. [14], CEA, France carried out 360 °C pressurized water (typical conditions of pressurized water reactor (PWR): [B] = 650 ppm, [Li] = 2 ppm, no addition of dissolved $O_2$ or $H_2$) static autoclave corrosion tests. The results indicate that although Cr coatings before optimization are not completely protective, they still reinforce the overall corrosion resistance of the sample compared to the uncoated Zr-4. Later, accelerated corrosion tests were carried out on the optimized Cr coated Zr-4 with the steam test condition at 415 °C and 100 bar (10 MPa), and the

corrosion was extended to 200–250 days [2,15]. The obtained test results are shown in Figure 3a. It can be seen from this figure that before optimization, gray stains are present on the surface of Cr-coated samples. This could be due to that Cr coatings without optimization possess local penetrating cracks, which lead to forming a $ZrO_2$ "Islet" below them. However, there is no sign of the acceleration of local corrosion, and a coating around the "Islet" does not break. The data showed that the weight gain of the uncoated Zr-4 is 260 mg/dm$^2$ after 200 days, forming a $ZrO_2$ layer of 17 μm. The weight gain of Cr-coated samples after optimization is less than 5 mg/dm$^2$, and a $Cr_2O_3$ layer of less than 100 nm is formed after 100 days. In terms of hydrogen absorption, the hydrogen content of uncoated Zr-4 reaches 1000 wt.ppm, while that of the Cr-coated samples before and after optimization are 150 and 25 wt.ppm, respectively, under the same conditions. This indicates that Cr coating effectively prevents hydrogen absorption of Zr alloys. Moreover, the results of water containing lithium (70 ppm Li) corrosion tests conducted by Bischoff et al. [16] (Figure 3b) showed that the uncoated M5 cladding tubes had undergone a "breakaway" corrosion after 140 days (turning curve), while the single-sided Cr-coated cladding tube still did not show any signs of "breakaway" until after 168 days. No dissolution or weight loss of the Cr coating was detected. This suggests that Cr coatings reduce the corrosion sensitivity of Zr alloys under the environments of lithium.

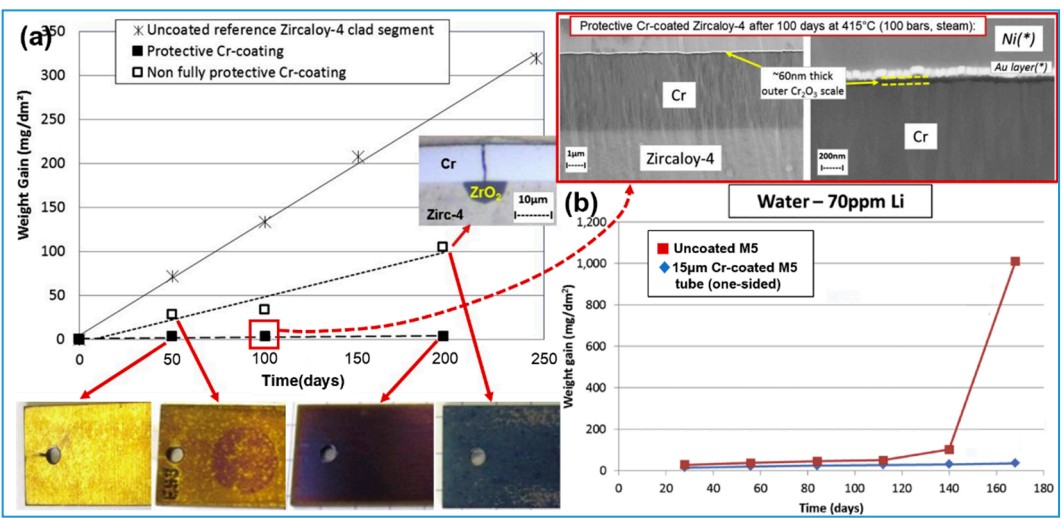

**Figure 3.** The results of corrosion test of Zr alloy cladding with PVD Cr coating. Reproduced from the work in [2] (**a**) and the work in [16] (**b**) with copyright permission from Elsevier.

Wei et al. [17] from the Nuclear Power Institute of China (NPIC) used static and loop autoclave to conduct corrosion tests of boron- and lithium-containing water ([$H_3BO_3$] = 200 mg/kg, [LiOH] = 1.2 mg/kg) and oxygen-containing water (dissolved oxygen [DO] = 100 ppb) on Cr-coated Zr alloys for 3000 h to simulate the hydro-chemical conditions for pressurized water reactor (PWR) and boiling water reactor (BWR), respectively. It was found that the corrosion properties of Cr-coated samples are much better than those of uncoated Zr-4, and the thickness of the oxide film of Cr-coated Zr alloys is only 50–100 nm after 3000 h. In contrast to PWR water, oxygen-containing water accelerates the corrosion of uncoated Zr-4 while reducing the corrosion rate of Cr-coated Zr-4. This could be due to the trace amount of DO in water, which might facilitate the formation of a thicker oxide film of Cr coatings and prevent their further oxidation. The preferential corrosion was also observed along the grain boundary on the surface of the Cr coating. It should be noted that although no significant dissolution was reported, an alteration in the color of the oxide film of the coating indicates a change in its valence state, suggesting that the oxide film may be slightly soluble. In contrast, the actual high-temperature and high-pressure coolant flush in the reactor may exacerbate this process.

Krejčí et al. [18] conducted single-sided (only the outer surface of the cladding was exposed after end plugs welded) long-term autoclave corrosion tests on Cr-coated Zr1Nb alloy cladding under

simulated water condition of water–water energetic reactors (WWERs). The results disclosed that the weight gain of Cr-coated samples is rather low, and the cumulative weight gain is almost constant after 100 days. However, the hydrogen content is only slightly lower than that of the uncoated samples (16 ppm after 210 days). This is somewhat different from the understanding that "the oxide film of the coated samples is thin and the hydrogen production is much lower". Brachet et al. [2] concluded that "Cr coating significantly improves the hydrogen absorption of Zr alloys". The authors proposed two possible explanations: First, thin $Cr_2O_3$ and metal Cr are less resistant to hydrogen permeability than protective $ZrO_2$. Second, the corrosion reaction of Cr results in higher absorbed hydrogen fraction [18]. However, these justifications have not been substantiated. Besides, the test results of 528-h in situ electrochemical corrosion impedance spectroscopy showed that the charge transfer resistance (corresponding to the reciprocal value of the corrosion rate) of the Cr-coated samples is 10 times higher than that of the uncoated samples, indicating an excellent protective behavior.

### 2.3. Fretting Wear Behavior

Under the influence of coolant flush and vibration, the potential fretting wear effect between fuel rods and grids in the nuclear fuel assembly as well as the debris is one of the significant mechanisms of fuel failure in light water reactors (LWRs) [19]. As the hardness of Cr is higher compared to Zr alloys, the application of Cr coatings is expected to improve the performance and data that have been obtained through different types of fretting wear tests.

Brachet et al. [2] conducted wear experiments on the uncoated and Cr-coated (6 μm) Zr-4 cladding samples under simulated chemical conditions of PWR primary cooling water. All the corresponding wear materials were uncoated Zr-4 sheets. The flow direction of the circulating water was parallel to that of the reciprocating movement of the wear drive system. The results displayed that under similar test conditions, the uncoated Zr-4 cladding showed noticeable wear, and the total wear depth of the wear Zr-4 sheet was 65 μm. However, the outer surface of the cladding with Cr coatings was not affected, while the wear depth of the wear Zr-4 sheet increased to 85 μm.

To simulate actual wear conditions, Bischoff et al. [16,20] and Delafoy et al. [19] carried out fretting wear tests in the loop by placing uncoated and Cr-coated M5 cladding samples in a grid cell that contained only one dimple or spring (Inconel alloy) for wear, respectively. The total 100 h tests were conducted under the chemical condition of PWR primary cooling water ([B] = 1000 ppm, [Li] = 2 ppm) at 300 °C. The obtained results as shown in Figure 4 reveal that for the cladding/dimple configuration, the wear volume of Cr-coated claddings is reduced by nearly 98% compared to the uncoated ones, and almost no visible wear could be observed. The dimple on the other side has more wear and gradually develops a cladding shape. In the cladding/spring configuration, the wear of the Cr-coated cladding and spring is negligible. In contrast, the uncoated cladding exhibited wear, and the total amount of wear of the Cr-coated cladding was reduced by two orders of magnitude.

They also used AISI316 stainless steel wire to simulate extremely severe debris wear [20]. The steel wire was fixed, and a sliding displacement of +/−200 m was applied to the cladding with a contact force of 1.5 N. After 23 h, the maximum wear depth of the uncoated sample was approximately 500 μm, which is almost equivalent to the entire cladding thickness. After 100 h, the maximum wear depth of the Cr-coated sample was ~280 μm, which is almost half of the former, and the notch on the sample was reduced in both size and depth. Even when the Cr coatings were fully penetrated, the wear rate remained low. They suggested that the steel wire could still in contact with the Cr coatings on the edge of the worn area.

Wen et al. [21] conducted fretting wear tests on conventional Zr cladding and Cr-coated Zr cladding in water and air environments at room temperature under normal and axial loads, respectively. It was found that the coating process exerted a significant impact on the fretting wear performance of the Cr-coated Zr cladding. Besides, the fretting wear rate was significantly reduced with water lubrication.

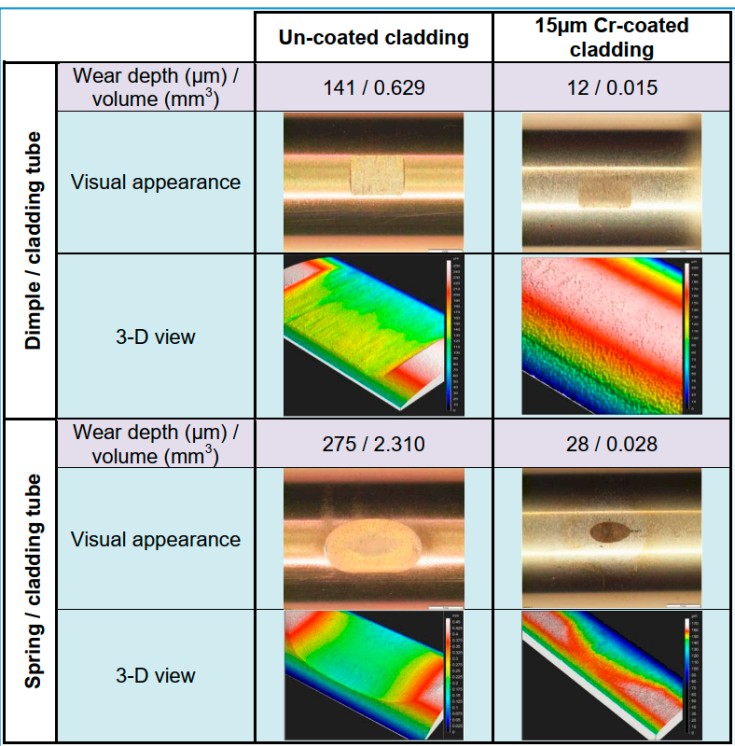

**Figure 4.** The results of wear test for uncoated and Cr-coated M5 alloys. Reproduced from the work in [19] with copyright permission from the American Nuclear Society (ANS) and the author.

### 2.4. Irradiation Behavior

In addition to high corrosion resistance, the irradiation stability of protective coating on the fuel cladding is not less than that of Zr alloy matrices. Unfortunately, neutron irradiation in the reactor is time-consuming and expensive, and it also demands special requirements of transportation, handling, and identification of radioactive samples. Conversely, ion irradiation can produce a high-dose in a short time, thereby reducing the irradiation cost and characterization.

Diverse methods to attain Cr coatings were also considered by few researchers. As elaborated in Part I of this review, Maier et al. [22] compared the ion-irradiation resistance of as-deposited and annealed CS Cr coatings and indicated that the preexisting deformation-induced defects in Cr coatings with severe plastic deformation act as a trap for resisting irradiation-induced damage. Moreover, Kuprin et al. [23] analyzed the irradiation resistance of arc-ion-plated (AIP) Cr coatings (5 μm) after irradiation using argon ions of 1.4 MeV to doses of 5–25 dpa (displacements per atom) at 400 °C. It was found that ion-beam irradiation induced the grain growth and isotropic increase of Cr coatings. At the irradiation doses of 5 dpa and 15 dpa, the average grain size of Cr coatings remained the same, while at a dose of 25 dpa, the grain size increased (<20%). However, at the same temperature, vacuum annealing without irradiation did not lead to increasing the grain size in the Cr coating. This implies that the irradiation-induced grain growth appears in a region that is independent of temperature. This could be due to the direct impact of the thermal spike on the grain boundary, i.e., the migration of grain boundary is caused by the atomic hopping in the thermal spike and is driven by the curvature of the local grain boundary. An irradiation-induced void was also formed in Cr coating, and with an increase in the irradiation dose, the size was increased while its concentration decreased. A void-free zone with a width of 3–5 nm was observed along the grain boundary, denoting vacancy swelling. The swelling degree of Cr coating was 0.16% at a dose of 5 dpa and 0.66% at a dose of 25 dpa, which is an order of magnitude lower than the allowable swelling degree of reactor core materials (~5%), signifying that Cr coatings possess excellent irradiation resistance.

Wu et al. [24] conducted ion irradiation tests using 20 MeV $Kr^{8+}$ on Cr-coated (2.5 μm) Zr alloy prepared by magnetron sputtering. The irradiation damage at the Cr/Zr interface was calculated to be 10 dpa. No significant change was noted in the concentration distribution of Cr and Zr before and after subjecting to irradiation (Figure 5a). Still, the concentration of Fe near the interface increased, and its concentration on the Zr side was higher compared to Cr side (Figure 5b). An analogy between heat-treated and irradiated samples confirmed that this was due to the effect of irradiation and not due to the effect of annealing (Figure 5c,d). It has been proposed that under irradiation, excess Fe at the interface occurs due to partial dissolution of the second-phase particles (SPPS) in the Zr-4 substrate near the interface, and in Zr the interstitial state is induced by ion irradiation at 400 °C, thereby augmenting the diffusion of Fe. After irradiation, the concentration distribution of Fe at different positions along the interface showed significant differences, indicating that Fe has preferential diffusion in grains with different orientations. HRTEM analysis exhibited that the original Laves phase C15 at the interface disappears after irradiation. This is due to the increase in the concentration of Fe near the interface during irradiation, which increases the electron-to-atom ratio (e/a), consequently stabilizing the C14 phase. However, the continuity of lattice matrix was maintained at the interface after irradiation, indicating that the Cr-Zr interface has excellent microstructure stability. Moreover, the tensile test of ion-irradiated coating samples revealed that the coatings had high residual adhesion.

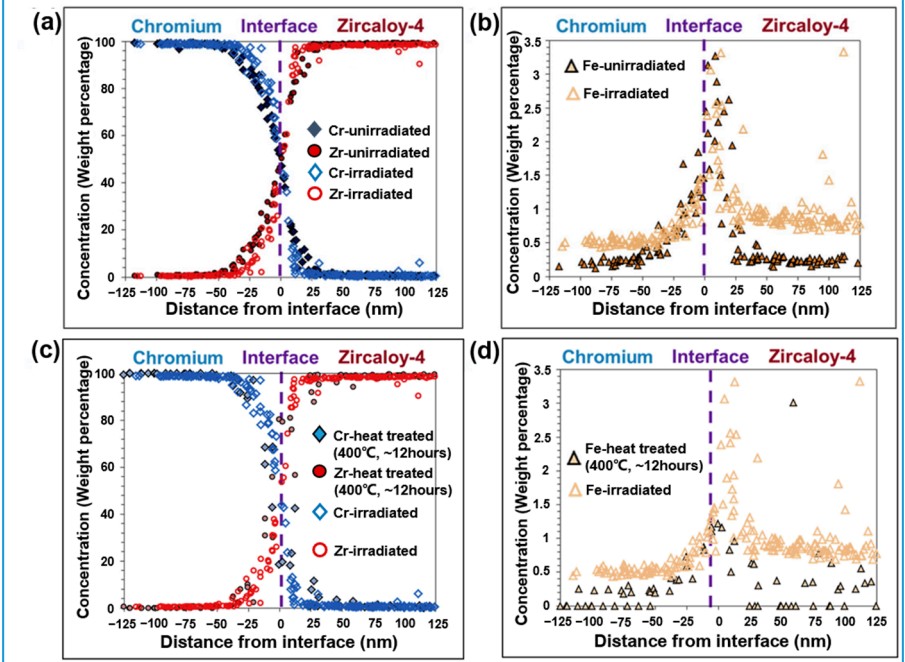

**Figure 5.** Impact of ion-irradiation on the chemical composition at the Cr/Zr interface: concentration distribution of Cr, Zr (**a**) and Fe (**b**) before and after irridiation; comparisons between heat-treated and irridiated samples in the concentration distribution of Cr, Zr (**c**) and Fe (**d**), respectively. Reproduced from the work in [24] with copyright permission from Elsevier.

Some researchers also have investigated the generation and evolution of cavities or bubbles induced by light ions (hydrogen and helium). Actually, in addition to inducing lattice damage, neutrons may also trigger transmutation reactions of target atoms, accompanied by the production of hydrogen and helium. For example, thousands of appm (atomic parts per million) helium will be induced in the zirconium alloy claddings by the impact neutrons after 54 months of PWR operation [25]. These light gas ions are prone to migrate, aggregate, and grow in the materials, causing performance degradation and volume swelling.

Huang et al. [25] carried out in situ irradiation tests at 400 °C with 30 kev He$^+$ on the cross-sectional samples of Cr-coated zirconium alloys prepared by the electroplating method. As the interlayer between the Cr coating and the Zr substrate, the Ni layer forms metallurgical bonding with the Zr matrix through annealing pretreatment to enhance the adhesion of the epitaxially growing Cr coating. It was found that helium bubbles distribute uniformly in the Cr coating when the helium concentration reached 28,900 appm (much higher than the helium concentration under real service conditions), but a bubble-free region was observed near the interface between the Cr coating and the Ni interlayer. It is believed that the interface between the coating and the transition layer has a high bonding quality due to the fact that He bubbles condensed at such a high helium concentration dose not destroy the interface. Besides, the size and density of He bubbles in the Cr coating were found to be increased with the helium concentration from several hundreds to tens of thousands of appm. Meanwhile, the average size of the bubbles formed at the grain boundaries was larger than that within the Cr grains, and the bubbles further aggregated to form short-bubble microcracks which may be considered as channels for oxidizing substances to reduce the corrosion/oxidation resistance of the Cr coating. However, an excellent resistance to the He$^+$ irradiation of the Ni interlayer was found to block the channels.

Jiang et al. [26] used multiple ion beams of 5 MeV Fe$^{2+}$, 2.9 MeV He$^{2+}$, and/or 270 KeV H$^+$ to irradiate $\alpha$-Cr at 475 °C. It was found that (Fe$^{2+}$ + He$^{2+}$) irradiation significantly reduced the sizes and increased the number density of cavities, compared with the single Fe$^{2+}$ irradiation. On the contrary, (Fe$^{2+}$ + H$^+$) irradiation resulted in a sharp increase in cavity size and a decrease in number density. However, (Fe$^{2+}$ + H$^+$ + He$^{2+}$) irradiation significantly increased the cavity density. The above results show that helium promotes the cavity nucleation, while hydrogen promotes the cavity growth, and they have synergistic effects on the cavity formation and evolution. More specifically, in the presence of helium, hydrogen also promotes cavity nucleation. In addition, the first-principles calculations revealed that the presence of He strengthens the hydrogen trapping effect of vacancies which stabilizes the initial embryos of cavities, thus accelerating their nucleation, providing an explanation to the mechanism of the synergy from the atomic model.

On the other hand, obvious material degradation phenomena under ion irradiation, such as the irradiation damage and amorphization, have been reported by some recent papers on Cr$_2$AlC [27,28] and SiC [29,30], which further confirms the superior performance of pure Cr as a candidate ATF coating material in terms of resisting against the irradiation damage and maintaining the crystal structure.

## 3. Performance under Accident Conditions

### 3.1. High-Temperature Oxidation Behavior

For the ATF case, the fundamental requirement is to decrease the oxidation rate of high-temperature steam inside the case, which will greatly reduce the generation of heat and hydrogen, making the emergency core cooling system (ECCS) during severe accidents less burdened. [31] A commonly employed method to evaluate the resistance of materials to reactions, such as high-temperature oxidation, is to characterize the kinetics of the related processes based on the relationship between oxidation weight gain, the thickness of the reaction layer, and metal consumption over time. The growth rate of a dense oxide layer is controlled by the diffusion of the reactants (metal and oxidant) through the oxide layer. According to Wagner's oxidation theory, the oxidation reaction with the diffusion process as a rate-controlling step can be described by a parabolic kinetic law, meaning that the oxidation rate is inversely proportional to the thickness of the oxide layer, X [32]:

$$\frac{dX}{dt} = \frac{k_p}{X}, X^2 = 2k_pt, \tag{1}$$

where $k_p$ is the parabolic rate constant. The rate constant measured using the weight gain can be obtained as follows,

$$\left(\frac{\Delta W}{A}\right)^2 = k'_p t, \tag{2}$$

where $\Delta W$ is the weight gain and $A$ is the surface area involved in the reaction. As it is more convenient to determine mass change than oxide layer thickness, most studies employ the weight gain curves to characterize the oxidation kinetics. The conversion relation between $k'_p$ and $k_p$ can be calculated based on the equilibrium relation of the metal oxidation reaction. Regardless of the temperature, the parabolic rate constant ($k_p$ or $k'_p$) is commonly reported as a unique oxidation rate constant. If the weight gain curve deviates from the parabolic law (i.e., transition), the degradation of the oxide layer may occur.

Numerous previously reported studies on high-temperature oxidation of coated Zr alloy claddings provided relevant kinetic data. However, these data are used for a direct comparison with uncoated Zr alloys, where only a few studies analyzed the oxidation kinetics data in detail. Yeom et al. [33] studied the oxidation kinetics of a Cr-coated Zr-4 plate sample prepared via cold spaying method within the temperature range from 1130 to 1330 °C up to 90 min. The obtained results showed a parabolic oxidation behavior at 1130 °C, with the rate constant $k_p$ of $3.6 \times 10^{-1}$ μm$^2$·min$^{-1}$. However, the kinetic data at 1230 and 1330 °C deviated from the parabolic behavior, and they were better described by the power law dependence of the oxide layer thickness on time ($X = k_n \cdot t^n$); the corresponding oxidation index, $n$, was approximately 0.25. Compared to the parabolic law ($n \sim 0.5$), the decrease in the $n$ value can be attributed to the influence of high-temperature conditions on the volatility of the oxide layer. Besides, the same study showed that the growth of the intermetallic layer at the Cr-Zr interface follows the parabolic law.

Brachet et al. [34] presented a parabolic kinetic curve and determined that the corresponding $k_p$ value in the temperature range from 800 to 1300 °C up to 300 s is consistent with the value obtained from the calculation with the self-diffusion coefficient of chromium oxide polycrystalline. However, based on these values, the authors considered that the high-temperature volatilization of Cr oxide made negligible contribution to the mass loss of the Cr coating given the experimental conditions. The results of the study were used as a base to evaluate the kinetics of the total consumption of the Cr coating and to establish a prediction model [2]. The model showed that the Cr coating with a thickness from 5 to 10 μm can withstand the oxidation under typical or slightly harsher design basis accident (DBA) conditions. Besides, they also analyzed the kinetic curve of one-sided steam oxidation of the Cr-coated M5 cladding, with the thickness of the Cr layer from 12 to 15 μm at 1200 °C for longer times (up to 12,000 s); the mass of the Cr-coating rapidly increased at 5000 s (Figure 6). This is related to the deterioration of the coating and loss of its protective properties, which yielded an increased oxygen intrusion [34,35].

A Zr-coated alloy is a multiphase system. Compared to a single-phase system, the phase interface and compositional gradient between different phases and the difference in their physical and chemical properties render the microstructure and the phase evolution during the oxidation more complex. Meanwhile, these microscopic evolutions may also reflect the oxidation degree, the transport of material, and possible degradation behavior of the entire system during the oxidation process. The microscopic evolutions were extensively studied since they may provide invaluable knowledge on the high-temperature oxidation behavior.

Vast studies reported that Cr-coated Zr alloys quickly transform into a typical four-layer phase structure after high-temperature oxidation at temperatures lower than 1300 °C, and the Cr coating is not completely consumed [17,33,36–39]. In the inward direction, the four existing phases in the order of occurrence are chromium oxide ($Cr_2O_3$); residual metallic Cr, $ZrCr_2$, or Zr (Cr, Fe)$_2$ intermetallic phase; and Zr alloy matrix (Figure 7). Among them, the $ZrCr_2$ intermetallic phase is a typical end product of the high-temperature diffusion coupling reaction, which is usually generated at the Cr/Zr interface. It can also precipitate outside of the Zr alloy and form dispersed particles due to the diffusion of Cr into the matrix. As the hardness of $ZrCr_2$ is higher than that of the Cr coating or the Za alloy matrix,

the intermetallic phase exhibits a brittle behavior. However, besides the common above-mentioned features, different coating preparation methods or different oxidation conditions result in specific behaviors: oxides are formed on the surface of a dense coating deposited by PVD and other methods. Cr coatings prepared by the air plasma spraying (APS) method contain a notable amount of pores or cracks, providing channels for oxygen intrusion, and thus yielding the oxides with a certain content inside the coatings [36]. Besides, various microdefects like pores in the $Cr_2O_3$ layer, voids, or gaps at the $Cr_2O_3$/Cr interface; dispersed micropores in the Cr coating; and Kirkendall type pores in the vicinity of the $ZrCr_2$ layer form during the oxidation process [33]. These microdefects correspond to different formation mechanisms, and their formation and evolution may influence the oxidation kinetics and degradation behavior of the system [40–43].

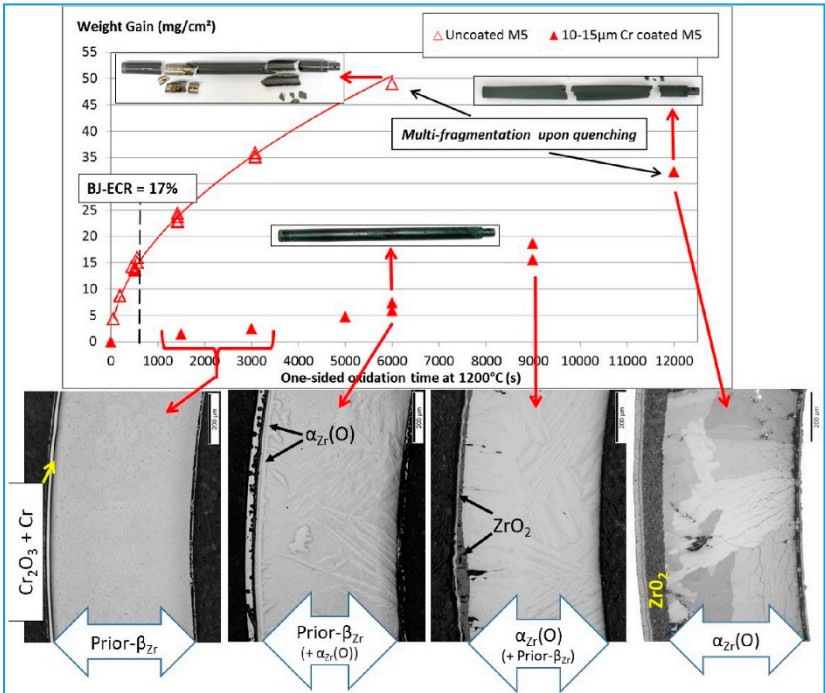

**Figure 6.** The kinetic curve of the one-sided steam oxidation of the Cr-coated M5 cladding rod at 1200 °C. Reproduced from the work in [35] with copyright permission from Elsevier.

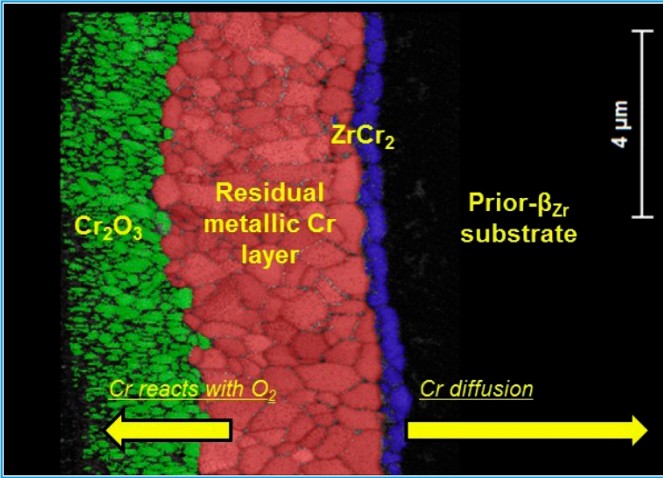

**Figure 7.** Phase evolution of the Cr-coated Zr alloy system during the high-temperature oxidation. Reproduced from the work in [2] with copyright permission from Elsevier.

The growth mechanism of $ZrCr_2$ at the coating/matrix interface is still highly debated. One assumption is that due to a quicker diffusion of Cr in Zr than that of Zr in Cr, the $ZrCr_2$ layer grows towards the Zr matrix, but the opposite behavior is also present [44]. The thermal-chemical stability of this intermetallic phase is also unclear. Kim et al. [3] and Park et al. [37] detected the mixed grains consisting of Zr-rich and Cr-rich phases at the Cr/Zr interface of the oxidized samples. The latter study brought a conclusion that these phases were formed during the cooling process due to the separation of the intermediate phases formed during the high-temperature oxidation. Recently, Brachet et al. [34] reported that the Zr (Cr, Fe)$_2$ layer grows steadily at the initial oxidation stage. However, when the oxygen diffusion occurred at a later step, the Zr (Cr, Fe)$_2$ layer disappeared and metallic Cr and $ZrO_2$ appeared as follows,

$$Zr(Cr)_2 + 2O \rightarrow ZrO_2 + 2Cr, \tag{3}$$

Besides, above 1300 °C, Cr coating and Zr matrix undergo Zr-ZrCr$_2$ eutectic reaction, and the exact eutectic temperature slightly varies among different samples, being in the range from 1305 to 1333 °C [34,45,46]. Brachet et al. [34] found that local oxidation kinetics of the eutectic and Cr-depleted regions were nearly the same, and they believed that the liquid Zr-Cr phase formed on the outer surface of Cr-coated cladding at ultra-high temperature (>1300 °C) would not significantly change the high-temperature oxidation mechanism and kinetics of zirconium alloy matrix. However, Krejčí et al. [18] reported a higher oxidation rate of the coated sample than the uncoated one beyond the temperature of eutectic point, and that the multilayer Cr-CrN coating can effectively block the diffusion of Cr into the Zr matrix and inhibit the formation of Cr-Zr eutectic by forming a ZrN interface layer. Anyway, it has been confirmed that Cr coatings can play a good role in protecting zirconium alloys from oxidation at ~1300 °C for a relatively long time [33,39,46]. In other words, the limiting temperature of the effective protection in a certain period provided by Cr coatings against high-temperature oxidation is ~1300 °C, which indicates that the application of the Cr coating is expected to increase the peak cladding temperature (PCT ≤ 1204 °C) required by the existing safety criterion by nearly 100 °C [31]. Although it is still unclear exactly how much benefit such an increase in PCT will bring to the reactor core safety, this is indeed a breakthrough as it provides tens of years of improvement in the accident resistance compared to the uncoated Zr alloy cladding.

Wei et al. [17] and Hu et al. [38] performed high-temperature air oxidation experiments of the Zr-4 plate coated with ~20 μm thick Cr layer prepared via arc ion plating (AIP). However, bubbles, bulges, as well as cracks and volatilization of materials in the oxidation layer can be observed on the surface, see Figure 8. Based on the experimental data, Wei et al. [17] explained the degradation mechanism of the sample, which can be summarized as follows. Once the $Cr_2O_3$ layer reaches a certain thickness, it undergoes buckling deformation due to growth stress, and separates from Cr to form small bubbles; meanwhile, the $Cr_2O_3$ layer forms a loose external layer due to the high-temperature volatilization, which reduces the protective effect of $Cr_2O_3$ to a certain extent; besides, oxygen will penetrate through the Cr coating and diffuse into the Zr matrix underneath the coating; as the oxygen content increases, the Zr phase near the surface area of the matrix shifts from $\beta_{Zr}$ to $\alpha_{Zr}$ (O) phase, resulting in a volume expansion; due to the mismatch of hardness and ductility between these two phases, the cracks will be generated at the interface between the $\alpha_{Zr}$ (O) layer and the $\beta_{Zr}$ layer; several interconnected cracks form the initial position of the bulge, and a large bulge is generated later due to the existing thermal stress which during the cooling process. Hu et al. [44] also discovered the Cr/Sn segregation in the failure region, and the Zr- and O-rich channels distributed along the columnar grain boundaries of the Cr coating close to the Cr/Zr interface. They claimed that these changes played an important role in the sample failure. Brachet et al. [34] detected $ZrO_2$ strips at the Cr-coated columnar grain boundaries near the interface of steam-oxidized Cr-coated samples. These strips acted as a short-circuit for oxygen diffusion and caused premature failure of the Cr coating. This is related to the Zr diffusion into the Cr coating.

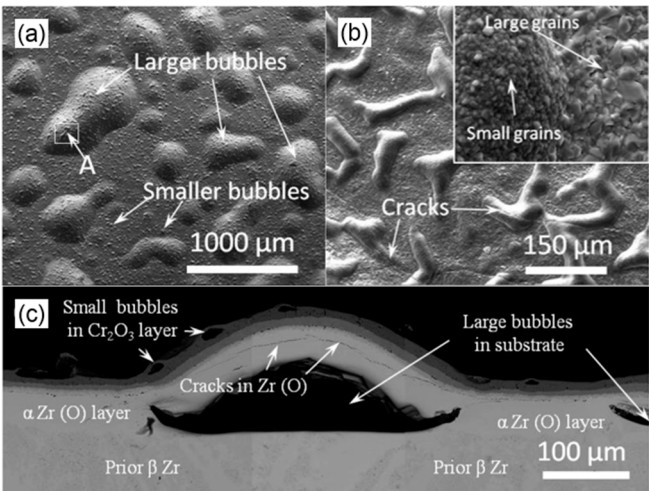

**Figure 8.** Degradation of the Cr-coated Zr-4 plate sample prepared via the arc-ion-plated (AIP) method after high-temperature air oxidation at 1200 °C (**c**): bubbles observed from the top-view (**a**) and side-view (**b**), respectively. Reproduced from the work in [17] with copyright permission from Elsevier.

Under LOCA conditions, cladding ballooning caused by internal pressure may result in macroscopic damage of the Cr coating, e.g., the formation of penetration cracks, but also different micro-defects, such as micro-pores, inclusions, and intercrystalline cracks. Therefore, the potential interplay between these coating defects and subsequent oxidation of the cladding is an attractive and important research question. French CEA made a series of surveys that opened primary discussions on this topic [2,15,47]. The studies showed that penetration cracks in the coating cause local oxidation of the Zr alloy matrix but not the acceleration of the Zr alloy oxidation. The existence of microdefects in the coating increases the coating degradation possibility and results in the separation of coating from the interface with the matrix, which can reduce the protective effect of the coating. Therefore, the control of the microscopic state of the coating is a plausible method to regulate the resistance of the coated Zr alloy to high-temperature oxidation, implying the importance of the proper understanding of key points of this approach.

Chaabane-Jebali et al. [48] investigated the effect of the preannealing heat treatment of the Zr alloy coated with a 6–8 μm thick Cr layer on high-temperature oxidation behavior. The Cr coating underwent recrystallization during the preannealing process, changing the grain morphology from initial elongated to more uniform grains. After the high-temperature oxidation, the rate of weight gain of the preannealed Cr coating sample was lower than that of the untreated sample, which was ascribed to the increase of the Cr grain size during the annealing, substantially lowering the subsequent oxidation and the amount of concomitant oxygen intrusion into the Zr matrix, see Figure 9. Based on the microhardness results, Yeom et al. [33] concluded that the residual Cr coating subjected to oxidation and cooling treatments was softer than its as-deposited state. This finding indirectly confirmed the occurrence of grain coarsening during the high-temperature oxidation. Brachet et al. [34] discussed these results from the point of preferential channels for oxygen diffusion that form in the grain boundaries during the medium-long high-temperature oxidation, which may be blocked by the annealing treatment (recrystallization of Cr), delaying the high-temperature oxidation. In the transient period upon the hypothetical accident, the nuclear fuel cladding may stay at a medium temperature (600 to 1000 °C) for a certain period before it reaches a relatively high PCT. Therefore, the Cr grain size (and morphology) may positively contribute to the high-temperature oxidation resistance of claddings when PCT is reached.

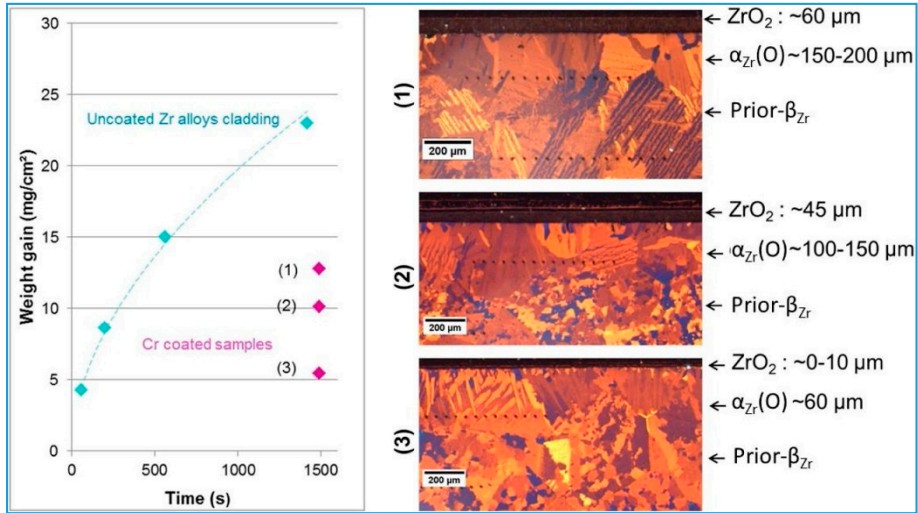

**Figure 9.** Effect of the pre-annealing treatment on the high-temperature oxidation behavior of the Cr-coated Zr alloy. Reproduced from the work in [34] with copyright permission from Elsevier.

Given that $Al_2O_3$ and $SiO_2$ exhibit better stability and better oxygen diffusion blocking than $Cr_2O_3$ at high temperatures, a few researchers investigated the high-temperature oxidation behavior of the Cr coating, but the obtained data were not reliable. Kim et al. [49] developed a CrAl model alloy, which exhibited better performance than a pure Cr alloy during the steam-oxidation at 1200 °C for 7200 s. Later, they prepared a CrAl coating on the surface of Zr alloy cladding using the AIP method, which dramatically improved the resistance to high-temperature steam oxidation compared to the uncoated Zr alloy [50]. Zhong et al. [51] prepared four different CrAl coatings (with Cr/Al atomic ratios of 41/58, 57/43, 67/33, and 81/19) on the Zr-2 plate using the magnetron sputtering (MS) method. The results of high-temperature steam oxidation at 700 °C for 20 h showed that a higher the Al content (including the pure Cr coating) decreases the oxidation kinetics. Wei et al. [52] and Dong et al. [53] prepared a $Cr_{1.63}Al$ coating and a $Cr_{62.8}Al_{27.9}Si_{9.3}$ coating using the multi-arc ion plating (MAIP) and the magnetron sputtering methods, respectively; however, the high-temperature oxidation resistance was not notably improved. Kim et al. [54] added 2 and 5% of Si to Cr-20% Al alloy. After exposure to steam oxidation at 1100 °C, it was shown that the higher Si content is related to the greater weight gain. Another feature of the alloy coating is that the alloying element will strongly interact with the Zr matrix, and its higher content will yield a higher amount of intermetallic phase [51]. Therefore, the reliability of the Cr alloy coating should be further investigated.

### 3.2. Embrittlement Behavior upon Quenching

The original idea of the proposed concept of coated ATF cladding is to effectively reduce the embrittlement of the zirconium alloy cladding caused by oxidation under accident conditions or to substantially extend the time until the cladding reaches full embrittlement under LOCA conditions, in that way giving more time to achieve the safety level of the reactor [1]. In the case of a hypothetical LOCA event, the 10CFR-50.46 regulations released by the US Nuclear Regulatory Commission (NRC) define safety standards for emergency core cooling systems (ECCS) [55]. Two of these standards—peak cladding temperature (PCT) limit of 1204 °C and the limit of maximum cladding oxidation of 17% (also known as the equivalent cladding reacted (ECR) limit)—are designed to prevent the fuel cladding from excessive embrittlement caused by high-temperature steam oxidation [35,56]. This embrittlement may weaken the ability of the system to cool down the core during and after the accident transient.

The ECR parameter is defined as the proportion of Zr alloy consumed relative to the initial thickness of the metal cladding. It assumes that the complete amount of oxygen that reacts with the cladding during the oxidation process and diffuses through the material will form stoichiometric $ZrO_2$.

For the calculation of the ECR parameter, most countries require the use of specific oxidation kinetics equations, which were derived from the steam oxidation experiment of Zr-4. Either the Baker–Just (BJ) equation [57], Equation (4), or the Cathcart–Pawel (CP) equation [58], Equation (5), can be used for the ECR calculations as follows,

$$W_{B-J} = 5770.6 \cdot t^{1/2} \cdot exp(-11450/T), \tag{4}$$

$$W_{C-P} = 602 \cdot t^{1/2} \cdot exp(-10050/T), \tag{5}$$

where $W$ (mg/cm$^2$) is the weight gain caused by oxidation, $T$ (K) is the oxidation temperature, and $t$(s) is the oxidation time. The ECR calculation considers the weight gain determined by the geometric factor of the cladding, and the factor which is based on the density and atomic mass of zirconium as well as on the molecular weight of oxygen. The ECR equations for the one-sided and two-sided oxidation are given as follows [56],

$$ECR_{\text{one-sided}} = 43.9[(W/h)/(1 - h/D_0)], \tag{6}$$

$$ECR_{\text{two-sided}} = 87.8(W/h), \tag{7}$$

where $W$ (g/cm$^2$) is the weight gain caused by oxidation, $h$ (cm) is the thickness, and $D_0$ (cm) is the outer diameter of the cladding. Depending on the type of oxidation of the Zr alloy, i.e., one-sided or two-sided, the ECR can be calculated using *B-J* and *C-P* equations or obtained from the weight gain data or the average oxygen content of the Zr matrix determined by EMPA [35]. As for the Zr alloy, the *B-J* equation was first used; however, the obtained results are more conservative at temperatures over 1000 °C, while the experimental data are more consistent with the C-P equation [35] under these conditions.

The ECR limit aims to ensure that the Zr alloy cladding retains ductility after high-temperature oxidation and quenching. The mechanical behavior of the cladding after high-temperature oxidation and quenching is usually evaluated by ring compression tests (RCT), which are performed at room temperature or 135 °C. The RCT test can give the estimation of apparent "residual ductility", defined as a displacement normalized to the outer diameter of the initial cladding measured in the moment of failure [56]. The limit of this parameter obtained from post-quenching (PQ) RCT is 1% or 2%, and it is used to define the macroscopic ductility-brittleness transition (DBT) of the PQ cladding [46]. Based on the above methods, various new Zr alloy claddings, such as ZIRLO, M5, HANA, and E110, were assessed against the ECR limits [59–63].

Recently, Krejčí et al. [45] extended the derivation method for the ECR limit of uncoated Zr alloy cladding to the evaluation of the residual ductility of Cr-coated E110 alloy cladding material (135 °C RCT) after one-sided high-temperature steam oxidation and quenching. The results were compared with the uncoated cladding materials. The coated samples did not exhibit DBT after being subjected to the oxidation at the following conditions; 1200 °C for 1 h, 1100 °C for 5 h, and 1000 °C for 48 h. However, as shown in Figure 10, when different ECR values were employed in the evaluation, a significant difference in the relationship between the RCT (%) of the coated cladding and ECR (%) was observed. Even if the weight gain remained at a very low level (3–5 ECR %), the coated sample was still brittle, not fulfilling the current safety standards. Therefore, the authors believe that Cr-coated materials can substantially improve accident tolerance, but a new safety standard is needed to evaluate the cladding fracture caused by local embrittlement. Similarly, Brachet et al. [35] reported that $ECR_{WG}$ used for uncoated cladding is no longer suitable for Cr-coated cladding because $ECR_{WG}$ has different physical meanings for the two materials (please see below). However, the B-J equation can be used to define new $ECR_{BJ}$ limits, which correspond to the performance of the Cr-coated cladding. The $ECR_{BJ}$ value that corresponds to the 1%–2% PQ RCT ductility limit of M5 cladding coated with a 12–15 μm thick Cr layer after one-sided steam oxidation at 1200 °C is approximately 50%, which is much higher than the current limit of 17% $ECR_{BJ}$ for Zr alloys. This indicates that the threshold time for PQ embrittlement of the Cr-coated M5 cladding is significantly extended.

Krejčí et al. [18] and Brachet et al. [35] discussed the PQ embrittlement mechanism of the Cr-coated cladding from the perspectives of hydrogen absorption, oxygen absorption, and element diffusion. Krejčí et al. [18] found that the PQ hydrogen content of all coated samples was very low, and the maximal value (120 ppm) is almost the same as for the uncoated sample. Therefore, they concluded that the DBT of the Zr matrix at lower experimental ECR values did not originate from the increased hydrogen content. Both research groups reported, similarly with the results for the uncoated Zr alloy cladding, that the PQ residual ductility of the Cr-coated cladding is in correlation with the average oxygen content of the residual "prior" $\beta_{Zr}$ layer. DBT occurs when the oxygen content exceeds the threshold of 0.4 wt.%. This indicates that the improvement of the PQ performance of the Cr-coated cladding is directly related to the ability of the Cr coating to delay the diffusion of oxygen into the Zr matrix during the high-temperature steam oxidation. Besides, Brachet et al. [35] found for the Cr-coated cladding with the DBT failure and low weight gain, the oxygen distribution between external oxide(s) ($ZrO_2$ and/or $Cr_2O_3$) and the metallic matrix does not match the typical distribution observed after the high-temperature steam oxidation of the Zr matrix cladding. This confirms the previously mentioned notions that $ECR_{WG}$ has a different physical meaning for the two materials.

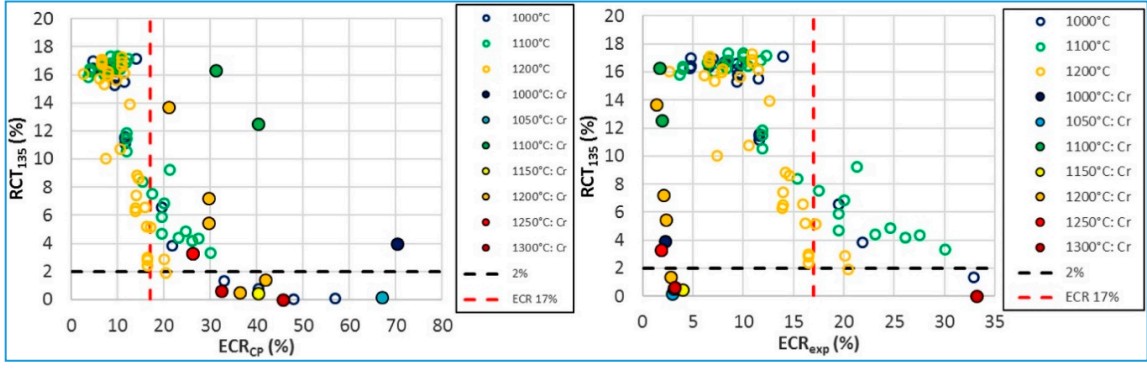

**Figure 10.** The relation between the post-quenching (PQ) residual ductility of Cr-coated E110 cladding (135 °C, RCT%) and $ECR_{CP}$ and $ECR_{EXP}$ indexes. Reproduced from [18] with copyright permission from Elsevier.

Regarding the Cr diffusion effect, Krejčí et al. [18] found that the hardness of the middle region of the cladding wall was higher than that in the inner region, as determined by the microhardness analysis. The difference in hardness was attributed to the Cr diffusion into the Zr matrix, which can contribute to the DBT enhancement. Furthermore, compared to the inert argon environment, the exposure to the steam environment accelerates the interaction between Cr and Zr. The authors suggested that Cr and O mutually promoted diffusion and migration to the "prior" $\beta_{Zr}$ phase, but the corresponding experimental evidence was not given. Brachet et al. [35] observed significant fluctuations in the Cr content along with different profiles in the Cr-diffusion direction (Figure 11) and speculated that these fluctuations were related to the distribution/fluctuation of Nb. As Cr exhibits a higher thermodynamic affinity to $\beta_{Zr}$ phase than Nb, during the growth and thickening process of $\alpha_{Zr}$ (O) at longer oxidation times, Cr tends to segregate in a thin residual "prior" $\beta_{Zr}$ region, leading to local hardening of the "prior" $\beta_{Zr}$ region. However, compared to oxygen diffusion, it has a secondary effect on the ductility of the cladding. This Cr/Nb segregation was similar to the Cr/Sn segregation reported by Hu et al. [38].

Another embrittlement phenomenon of the cladding sample can be observed after the high-temperature steam oxidation for a certain period when the fracture failure occurs upon the direct quenching in water. Brachet et al. [35] determined the times required for this critical oxidation at 1200 °C: for uncoated M5 cladding sample and the sample of M5 cladding coated with a 12–15 μm thick Cr layer, the critical time ranges are 3000 to 6000 s (38% < $ECR_{BJ}$ < 53%) and 9000 to 12,000 s (65% < $ECR_{BJ}$ < 75%), respectively. This confirms that the external Cr coating effectively improves the performance of the cladding exposed to high-temperature oxidation for the same time, making it able to withstand

final quenching, and providing the additional time for the total process, from high-temperature oxidation to the final structural failure during quenching. However, it has to be noted that even if the PQ residual ductility of the cladding declines to a low level after the high-temperature oxidation, it does not mean that the sample cannot withstand the water quenching. Kim et al. [50] performed high-temperature steam oxidation and quenching of the uncoated and CrAl-coated Zr-4 cladding samples. The experimental results showed that the CrAl-coated Zr alloy cladding maintained its integrity after the exposure to the temperature of 1200 °C for 3000 s, while the uncoated cladding was severely damaged by the thermal shock. Nevertheless, it should be noted that they first cooled the samples to 800 °C after the oxidation, and then, lowered the temperature to room temperature by water quenching. As these conditions are not as harsh as a direct cooling to room temperature by water quenching, more conservative evaluation results were obtained when the direct water quenching was employed.

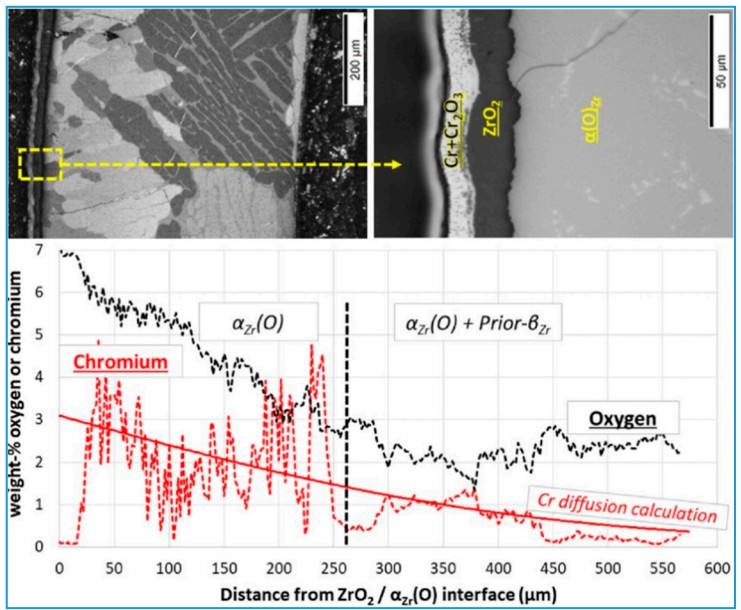

**Figure 11.** Cr diffusion profiles in the M5 PQ cladding coated with a 12–15 μm thick Cr layer after the high-temperature steam oxidation at 1200 °C for 9000 s. Reproduced from the work in [35] with copyright permission from Elsevier.

The temporal evolution of the embrittlement of the coated cladding is related to the evolution of the protective effect of the Cr coating. As mentioned in Section 3.1, after the exposure to high-temperature oxidation for a longer period, the kinetic curve of the Cr-M5 cladding entered a quicker alteration stage. According to Brachet et al. [35], the time required to fully protect the alloy from oxidation at 1200 °C for Cr coatings with a thickness of 12 to 15 μm is at least 3000 s, which is equivalent to an approximate $ECR_{BJ}$ of at least 40%. However, this time interval gradually shortened with the decline of the coating thickness. The ductility of the PQ coated cladding in this stage was hardly affected. At longer oxidation times (such as 9000 s), the coating gradually degraded and allowed for oxygen diffusion into the metallic matrix underneath the coating, which first caused the thickening of the $\alpha_{Zr}$ (O) layer, and then the formation of $ZrO_2$ layer underneath the coating. It should be noted that the residual unoxidized metallic Cr coating can be still detected at this time point, indicating that the Cr coating failed before complete oxidation of the coating. At this stage, a large proportion of the metal matrix underneath the coating was transformed into $\alpha_{Zr}$ (O), which yielded macroscopically brittle behavior of the PQ cladding at the room temperature. Besides, they found that transient hydrogen absorption occurred when the coating completely lost its protective effect [35]. The authors claimed that this is related to the formation of a discontinuous $ZrO_2$ intrusion region in the Cr coating. This phenomenon occurs at

high ECR values (50% to 60% for uncoated materials), and hydrogen absorption at a level of hundreds of ppm (wt.%), reaching a saturated hydrogen content after a short transient period. It was further pointed out that the protection provided by the Cr coating during the high-temperature oxidation process depends on many parameters, especially the thickness of the coating and the deposition method on the matrix. Besides, the ballooning/bursting and internal oxidation that occurred during the LOCA transient should be also considered.

Park et al. [64] performed four-point bending and RCTs on Cr-coated and uncoated Zr alloy claddings after the simulated LOCA transient ballooning/bursting to further evaluate the impact of the Cr coating on the mechanical properties of the cladding. Among these tests, the RCTs were employed to evaluate the ductility at equal distances from the ballooned zone. The obtained results can be summarized as follows.

- In the four-point bending tests, the failure of all samples (including coated samples) occurred near the center of the rupture region. The authors concluded that the failure position in the rupture region is determined by the high-degree local oxidation and the decrease in the wall thickness caused by ballooning. On the other hand, the steam that remained in the ballooned zone caused severe hydrogen absorption near the rupture open, transforming these brittle regions with high hydrogen contents into failure positions.
- The maximum load of the Cr-coated Zr alloy cladding is higher than that of the uncoated sample. This may originate from a higher average wall thickness of the Cr-coated sample, where the oxidation of the outer surface does not occur to a significant extent.
- The wall thickness of the middle-plane sections within the ballooned zone greatly varies, yielding different oxidation rates at different circumferential positions in the same moment of the oxidation process. The position edge of the rupture has the lowest wall thickness and the highest oxidation degree. The cracking starts from the end of the brittle rupture opening and quickly expands through the ballooned zone. Therefore, the bending strength of the bursting tube mainly depends on the thickness of the load-bearing Zr matrix in the cladding wall section opposite to the rupture.

The strain derivations of the Cr-coated sample and uncoated control were 4.7% and 1.67%, respectively, indicating that two samples exhibit plastic and brittle behaviors, respectively. This also confirms that the Cr-coated sample retains ductility for a longer period.

## 3.3. Ballooning/Bursting Behavior during LOCA Transient

It is well established that the Zr alloy cladding can withstand the coupling effect of high-temperature steam and internal pressure in the early transient stage of LOCA conditions. The surface of the Zr alloy cladding is continuously oxidized by the high-temperature steam, yielding continual degradation of the respective mechanical properties; on the other hand, the outside pressure around the cladding decreases due to the loss of coolant, while the internal pressure of the cladding rises due to overheating. Therefore, the cladding undergoes plastic deformation due to this large pressure difference. Specifically, the synergistic effect is that the Zr alloy cladding balloons and bursts within a temperature range from 700 to 1000 °C [59]. The ballooning will affect the cooling capacity of the fuel assembly, while bursting will cause oxidation and secondary hydriding of the inner surface of the region near the cladding tube rupture. Therefore, it is necessary to investigate the ability of the cladding coating to maintain its integrity during the ballooning/bursting processes, as well as the potential influence of the coating on the high-temperature mechanical properties of the cladding. At present, there are only a few reports and research activities on this topic.

Kim et al. [3] conducted a simulated LOCA ramp ballooning/bursting test on a 200 mm long Cr-coated Zr-4 cladding (the length of the coated region was 100 mm) under the conditions of hoop stress of 35.25 MPa and a heating rate of 14 °C·s$^{-1}$. The results showed that during the temperature increase to 1200 °C, all cladding samples ballooned and burst, and the ranges of bursting temperature

of Cr-coated and uncoated cladding samples from 941 to 972 °C are similar under conditions of similar heating rate and internal pressure. However, the bursting of the partially Cr-coated sample occurs is in the uncoated region, indicating that the bursting resistance of the Cr-coated cladding is better than that of the uncoated sample. The increase of the bursting resistance of the Cr-coated area (coating thickness of ~80 μm) may originate from both the increase in the cladding thickness in this area and the improvement of the overall bursting resistance of the cladding due to the Cr coating prepared by 3D LMC technology. These findings are consistent with the results of previous ring hoop tension/ring compression tests [3].

Brachet et al. [65,66] performed an isothermal internal pressure creep bursting test in steam on the Cr-coated Zr-4/M5 cladding (coating thickness of 10 to 15 μm) with a gauge length of 300 mm. An uncoated sample was used as a control. The test temperature covers the main range of bursting temperatures under DBA LOCA conditions, from 600 (α-phase region) to 1000 °C (β-phase region). Their findings are summarized as follows.

- Under the given internal pressure, both Cr-coated Zr-4 and M5 claddings exhibit the creep rupture time 2 to 3 times higher than that of the uncoated samples.
- When the creep temperature is below 850 °C (α-phase region), the average and maximum circumferential strains of the Cr-coated claddings are generally lower than of the uncoated samples; above 850 °C (α + β phase mixed area and β-phase area), although the Cr-coated cladding expanded significantly, the rupture size is very small (approximately 1 mm$^2$).
- Under the test conditions, the Cr coating always fully adheres to the substrate after ballooning/bursting (including the region in the vicinity of highly-deformed ruptures).

Based on the above findings, the authors concluded that the Cr coating imparts a high-temperature strengthening effect on the entire coated cladding. However, this strengthening effect is not caused by the increase in the thickness of the cladding wall but rather by the intrinsic strengthening effect of the Cr coating.

To get a better insight in the actual LOCA transient, Dumerval et al. [67] conducted further thermal ramp investigations on the Cr-coated Zr alloy cladding, and the main findings can be summarized and presented in Figure 12:

- For all samples, the bursting temperatures under high internal pressure (100 bar) and low internal pressure (10 bar) conditions are within the range of 700 to 800 °C ($\alpha_{Zr}$ phase area) and above 1000 °C ($\beta_{Zr}$ phase area). Besides, the bursting and temperature of the Cr-coated Zr alloy cladding are, under most circumstances, higher than those of the corresponding uncoated samples tested at a similar heating rate.
- Under the special conditions of low internal pressure (10 bar) and a high heating rate (25 °C·s$^{-1}$), only limited creep deformation without bursting was observed on the Cr-coated M5 cladding when the temperature reached 1133 °C; the bursting temperature of the uncoated sample is in the range of 1030 to 1090 °C (β phase region). This indicates that the introduction of Cr coating increases the bursting temperature by at least 50 °C.
- The axial ballooning range and circumferential elongation of the Cr-coated M5 cladding are substantially lower than those of uncoated M5.
- When the heating rate is higher or equal to 1 °C·s$^{-1}$ (representing most typical LOCA transients), the "average" circumferential elongation($\varepsilon_{\theta\theta, h}$) of the Cr-coated M5 cladding is consistently lower than 30%, while the maximum circumferential elongation ($A_t$) at the bursting position is lower than 70%. The authors recommended that these measured values should be employed as the upper threshold of the cladding's behavior under LOCA conditions.

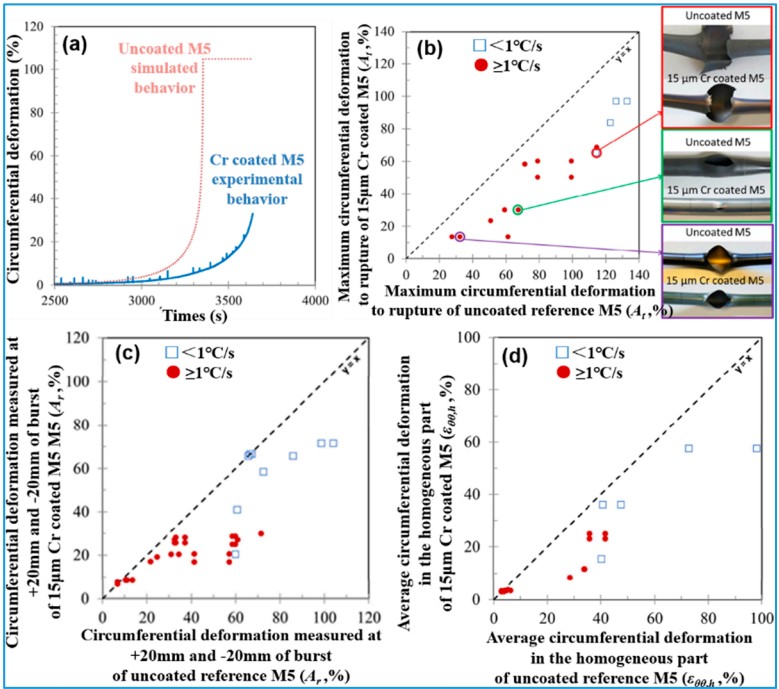

**Figure 12.** Results of the thermal ramp test on Cr-coated zirconium alloy cladding: (**a**) circumferential deformation of Cr-coated (experimental) and uncoated (simulated) M5 tubes as a founction of time; comparisons in the measured circumferential deformations — $A_t$ (**b**), $A_r$ (**c**) and $\varepsilon_{\theta\theta, h}$ (**d**), respectively — between Cr-coated and uncoated (both experimental) M5 tubes tested for different heating rates. Reproduced from the work in [67] with copyright permission from the American Nuclear Society (ANS) and the author.

Park et al. [64] conducted an integrated LOCA test on rod-shaped uncoated and Cr-coated (coating thickness of 20 to 30 µm) Zr alloy cladding samples with a length of 400 mm. Al pellets were loaded into the cladding to simulate the heat capacity of the fuel. The temperature was raised to 1200 °C (at a heating rate of 5 °C·s$^{-1}$) under internal pressure of 8 MPa and flowing steam. After dwelling at 1200 °C for 300 s, the samples were slowly cooled down and quenched at 800 °C. The test results are presented in Figure 13. The main findings are summarized as follows.

- The bursting temperature of the Cr-coated and uncoated Zr claddings are 839.5 and 754.8 °C, respectively.
- The balloon degree and the rupture size of the uncoated sample are larger than those of the Cr-coated sample.
- The circumferential strain of the middle-plane bursting of the Cr-coated and uncoated Zr cladding samples are 115.91% and 123.18%, respectively. The higher the circumferential strain, the lower the wall thickness after the bursting failure, and the wall has the lowest thickness near the bursting opening. Besides, oxidation occurs only locally on the inner surface of the ballooned and ruptured region, while the entire outer surface of the uncoated sample was oxidized. The former has a higher residual Zr matrix thickness (521 µm) than the latter (431 µm).
- Both samples exhibit a higher outer diameter strain along the direction parallel to the rupture than along the direction vertical to the rupture, while the Cr-coated sample has a much smaller maximal outer diameter strain and axial expansion zone.
- Although there are small axial surface cracks near the expansion section of the Cr-coated tube, cracking or spalling does not occur in the Cr coating to a significant extent. This indicates that despite the difference between the CTE of Cr and Zr, the rapid temperature change during the LOCA test still has a little effect on the bonding force between the Cr coating and the Zr matrix.

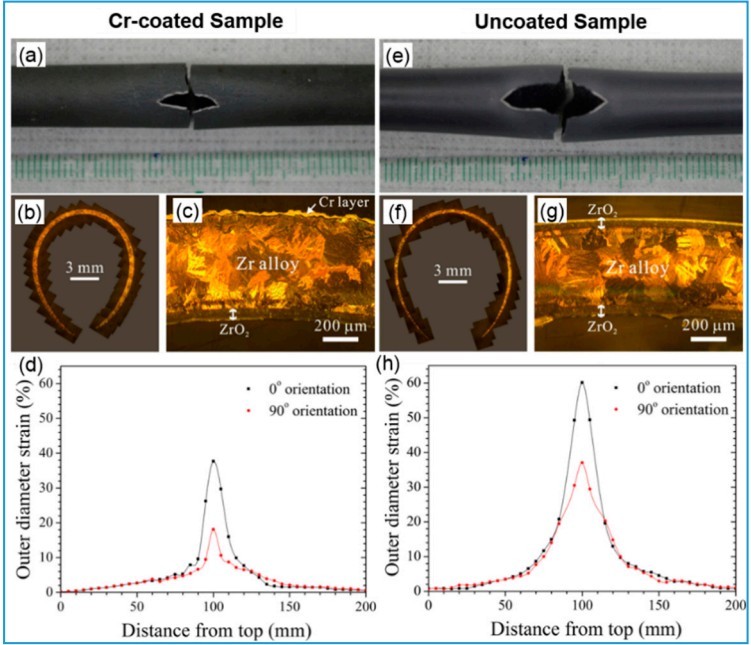

**Figure 13.** Results of the comprehensive LOCA test on the single-rod Cr-coated Zr alloy cladding (containing Al pellet): comparsions in the rupture mophology (**a,e**), residual cladding thickness (**b,c,f,g**) and outer diameter strain (**d,h**), respectively, between Cr-coated and uncoated samples. Reproduced from the work in [64] with copyright permission from Elsevier.

As mentioned in the first section of the review, KAERI proposed an ATF surface-modified cladding concept featuring integrated CrAl coating and the oxide dispersion strengthening (ODS) surface treatment. Kim et al. [50] evaluated the accident resistance of this modified cladding via a comprehensive LOCA test under experimental conditions similar to those reported by Park et al. [64] with an exception that the applied hoop stress reached 62 MPa. The results are presented in Figure 14 and can be summarized as follows.

- Compared to the uncoated Zr-4 cladding, the bursting temperature of the CrAl-coated cladding (coating thickness of 50 μm) or the sample with the ODS layer (thickness of 100 μm) was improved. Besides, the sample coated with both the CrAl layer and the ODS layer had the highest bursting temperature.
- The application of the CrAl coating greatly reduced the circumferential deformation of the cladding, and this was even further reduced by the ODS treatment. However, these techniques still failed to completely prevent the cladding from the bursting.
- No severe oxidation of the CrAl coating occurred on the outer surface of the Zr-4 cladding coated with the CrAl and the ODS layers, and the Zr matrix with the ODS layer did not react with the CrAl coating. Besides, no severe oxidation of the CrAl coating or ODS treatment layer occurred in the ruptured section.

Chalupová et al. [68] compared the isothermal bursting performance of three coated E110 claddings (Cr, multi-component CrN + Cr, and multilayer CrN/Cr, with a thickness within the 15 to 20 μm range) with uncoated control in an argon atmosphere. The testing was conducted within a temperature range of 600 to 1100 °C and in the range of internal pressure from 1 to 10 MPa. The main findings are given as follows.

- The bursting time and temperature of all coated samples were higher than those of the uncoated samples.

- The maximal deformation and axial deformation of all coated samples were significantly lower than those of the uncoated samples. Among the coated samples, Cr-coated cladding exhibited the highest deformation decrease, and the axial deformation of the cladding coated with multicomponent CrN + Cr is close to that of the uncoated cladding.
- At 800 °C, the rupture size of most coated samples was larger than that of the uncoated samples, which is contrary to the above-mentioned experimental results.
- Two failure mechanisms were observed near the ruptures of the coated samples: sparsely distributed long cracks and a large rupture appeared on the Cr-coated sample, while the CrN + Cr-coated sample exhibited densely distributed cracks and a smaller rupture.
- At 800 °C, the creep rate shifts rapidly and discontinuously with the change of ring hoop stress. When the ring hoop stress exceeds 40 MPa, the bursting time decreases, and a part of the Zr alloy remains as the α-phase, resulting in a low creep rate; when the ring hoop stress is lower than 40 MPa, the bursting time prolongs, and the zirconium alloy is converted into the β-phase, yielding a high creep rate.

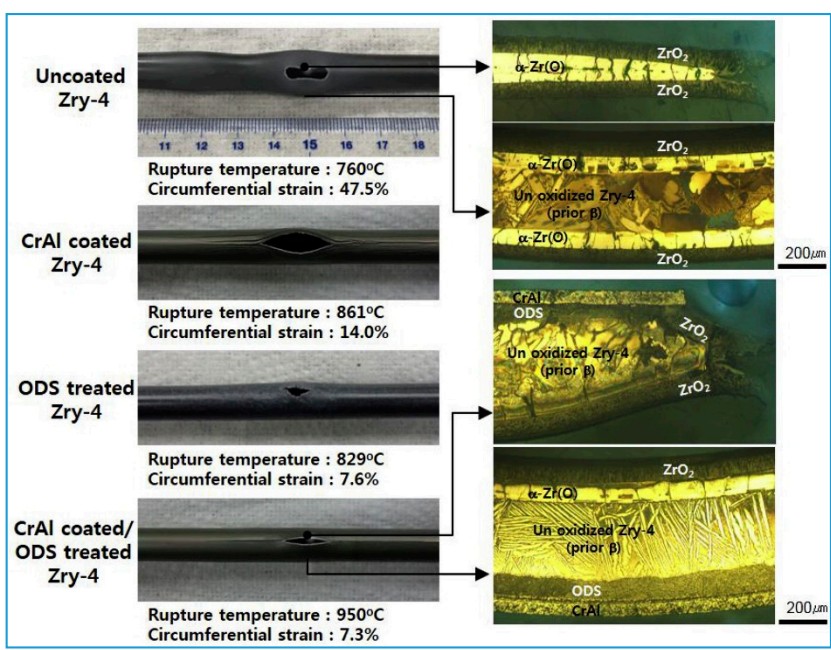

**Figure 14.** The comprehensive LOCA test on the Zr cladding coated with both the CrAl layer and the ODS layer. Reproduced from the work in [50] with copyright permission from Elsevier.

The reason for the formation of larger-sized rupture in the coated cladding is explained as follows. At a certain cladding deformation level, the coating cracked, locally increasing the creep rate; the thickness of the coated cladding in the cracked area declined, which caused stress concentration. During the subsequent bursting, the cladding burst opened along the entire crack length. The authors claimed that this would make a negative impact on the cladding. The hydrogen content of the matrix was increased due to the oxidation of the crack bottom, while the local mechanical properties were degraded as Cr in the crack region diffused into the matrix.

## 4. Discuss and Comment

Some issues regarding the preparation of the Cr-based coatings were mentioned in Part I of the review [1]. Herein, we discuss only the issues that occur during the research of the performance of Zr alloys with Cr-based coatings.

Accident resistance is the central point of the ATF concept and extensive research was performed on this topic. Particular attention was paid to the effect of Cr-based coatings on the improvement of

the comprehensive high-temperature oxidation resistance of Zr alloy claddings [2,3,33,34,36–39,44–54] and the mechanical properties (ballooning/bursting and embrittlement upon quenching after the oxidation) [18,35,64–68]. According to the available literature studies, the main mechanism of this improvement can be summarized as follows. Cr-based coatings physically prevent the contact between the Zr matrix and steam, directly avoiding any severe reactions; on the other hand, the formation of a protective dense oxide layer effectively suppresses diffusion of oxygen and hydrogen into the Zr matrix, yielding the substantial reduction or delay of the DBT of the Zr alloy. Therefore, the key point in the concept of the ATF-coated cladding is the quality of the coating. The primary prerequisite for the high-quality coating is bonding between the coating and matrix. Given that good performances of Cr-based coating on Zr alloys were achieved, it is reasonable to consider that the existing preparation methods can ensure sufficient bonding force. Therefore, effective protection relies on the consumption rate and degradation degree of the coating during the service life. The current state of the research can be summarized as follows.

(1) There are three pathways by which Cr-based coatings can be consumed: formation of surface oxide layer, volatilization of $Cr_2O_3$, and diffusion of Cr and other (impurity) elements or alloy elements (for the Cr-alloy coating) into the Zr matrix or reaction with Zr, generating an intermetallic layer. Under relatively short oxidation times, these two pathways have very limited influence on the coating, meaning that the consumption rate of the coating mainly depends on the formation rate of the surface oxide layer or, more precisely, on the oxidation kinetics [2]. However, there is little data regarding this topic to date to support any accurate evaluation. This is particularly the case for longer oxidation times. Besides, the research on the factors which can affect the oxidation kinetics of the coating is still not complete. The data exist only for the temperature influence [46], grain structure of the coating [48], and the chemical composition of an alloy (Cr alloy coating) [51]. Any further factors of influence, such as impurities in the coating, surface conditions, and atmospheric conditions, are not fully understood, and their effects and related mechanisms were not systematically studied.

(2) For the degradation of the Cr-based coating, current literature data have confirmed that the $Cr_2O_3$ layer formed on the surface of the Cr-based coating, and the coating itself, cannot efficiently prevent the oxygen diffusion [34]. Under short oxidation times, the coating is fully protective; however, during a long oxidation process, the coating gradually loses its protective properties, and oxygen diffuses into the Zr matrix through the entire coating, yielding the formation of the oxygen-rich $\alpha_{Zr}$ (O) brittle layer; when the coating completely loses its protective effect, a $ZrO_2$ layer grows on the surface of the Zr matrix underneath the coating. The degradation of the coating triggers the deterioration of the mechanical properties, including the bulging of the sample surface observed for the air oxidation [17,38] and the decrease in the PQ ductility, detected after the steam oxidation [18,35]. Factors that affect the degradation process of the coating include the deposition process of the coating, the thickness of the coating, and the diffusion of Zr into the Cr coating. However, we still lack systematic and in-depth research on this topic, which prevents us from fully understanding the interplay and mechanism of coating degradation under the influence of the above-mentioned factors.

(3) The existing research relies on the traditional method used to evaluate the embrittlement of the uncoated Zr alloy [15,18,35,45]. More specifically, the PQ embrittlement threshold (reflected by the ECR value) is employed to evaluate the protective effect of the coating to maintain the integrity of the cladding under accident conditions. However, compared to the uncoated cladding, the PQ embrittlement threshold of the coated cladding depends mainly on the oxygen content of the residual "prior" β layer; the improvement of the PQ performance of the coated cladding is directly related to the ability of the coating to delay the diffusion of oxygen into the Zr matrix during the high-temperature oxidation process (under typical DBA conditions). Therefore, experimentally used $ECR_{WG}$ to obtain the weight gain data for uncoated claddings is no longer suitable for coated claddings [18,35]. A potential solution of this issue is to use the calculation equation (*B-J* or *C-P*) to define the new ECR limit, which corresponds to the coated cladding performance; another solution is to reestablish the oxidation kinetics equation, which would be only suitable for coating materials and employ this equation to define new limits to

prevent embrittlement of coated cladding under LOCA conditions [35]. However, as mentioned above, the protective performance of the coating depends on many factors, especially the thickness and the deposition process of the coating. Therefore, this new equation may differ between different coating materials, implying that a new standardized method is required. Besides, the ballooning, bursting, and internal oxidation that occur during the LOCA transient need be taken into consideration. For the embrittlement mechanism of the coated cladding, besides the oxygen content and oxygen distribution, possible additional mechanisms include hydrogen absorption and the interaction between Cr and Zr matrix. However, the exact contribution of these factors to the embrittlement of the coated cladding remains unclear.

(4) The ballooning/bursting behavior of the cladding during the LOCA transient is an inherent feature of the Zr alloy cladding [59]. Several studies reported relatively consistent findings on the effect of Cr-based coatings on this process. Regardless of the testing conditions, Cr-based coatings improve bursting temperature and bursting time thresholds, and reduce the deformation degree [64–68]. However, the above thresholds differ between different coating materials or testing conditions, indicating that the creep rate of the coated cladding depends on these conditions. Therefore, it is necessary to clarify the internal relationship between them. There are two possible scenarios regarding the size of the rupture. First, the rupture size of the coated cladding is smaller than that of the uncoated cladding, which can originate from the improved high-temperature strength of the coated cladding [64–67]; second, the rupture size of the coated cladding is larger than for the uncoated cladding, which can originate from the cracking of the coating which causes local embrittlement and stress concentration, jointly generating a tearing effect [68]. Therefore, further research on the bursting failure behavior of the coated cladding is needed to understand the effect of coating cracking on the rupture size. As ruptures can cause internal oxidation and secondary hydrogen absorption, their impact on the ability of the coated cladding to maintain its integrity after quenching needs to be further evaluated.

Although there is a profound understanding of the accident-resistant behavior of Cr-based Zr alloy coatings, the investigation of some potential issues under normal working conditions seems to be overlooked. These issues are critical to maintaining the normal service status of the coating. It is already known that the coating is more prone to cracking than the Zr alloy matrix, and the crack will expand into the matrix [12,13]. The cracking behavior of the coating under different environments and loading conditions, or during the deformation of the matrix (e.g., fatigue, creep, etc.), as well as the effect of cracking behavior on the long-term structural integrity of Zr alloy claddings, remain unresolved. Due to differences in electrochemical activity and corrosion potential between the coating material and the Zr alloy, the electrochemical corrosion system can be formed at the interface [5]. The generation of the defects in the coating, for any reason, will cause a partial exposure of the interface, accelerating the interfacial corrosion of the Zr alloy. Despite these concerns, in-depth research on the electrochemical corrosion behavior of Zr alloy with Cr-based coating has not been conducted so far. Besides, there is still no definite agreement on the hydrogen absorption behavior of Cr-coated Zr alloys. Some studies suggested that a Cr coating greatly reduces the hydrogen absorption of a Zr alloy under normal conditions [2,14], while others did not find any substantial improvements [18,45]. In summary, there are still obvious gaps in understanding of the performance of Zr alloys with Cr-based coating under normal operating conditions, while comprehensive and systematic research under extreme working conditions is still lacking. To evaluate the service performance of the Cr coating in the reactor conditions more accurately, more systematic studies on the above issues are still needed.

One more issue worth discussing that has been in argument in the literature is whether or not to zirconium alloys should be coated with materials that can react with the substrate at high temperature. As far as we can see, it is a common physical process that the elements of the coating and the substrate diffuse along the concentration gradient to the direction of lower element concentration at high temperature. Due to the high concentration of elements at the coating/substrate interface, it is easy to form intermetallics by reaction diffusion. At present, this phenomenon has been reported in

many researches on ATF coatings, such as Cr, CrAl, FeCrAl, CrAlSi, TiAlN, TiAlCrN, $Cr_2AlC$, $Ti_2AlC$, etc. [2,51,53,69–73]. Of course, the diffusion of metal elements in ceramic coatings to the substrate is not as strong as that of the metal coatings, because the covalent bonds of ceramic compounds are much stronger than the metal bonds of pure metals or alloys. However, C or N elements diffuse faster than metal atoms in carbides or nitrides coatings into the matrix due to their small atomic size, and they also react with the Zr matrix at the interface to form ZrC or ZrN [45,73].

It is almost certain that this phenomenon exists in any of the currently proposed candidate coating materials for ATF, especially for those containing Fe, Al, and Si due to their high activity and high diffusion rate in the Zr matrix [51,69,70]. On the one hand, the mechanical properties of the coating/substrate interface or the whole cladding will deteriorate due to the formation of a large number of brittle intermetallics [51,74]. On the other hand, the rapid loss of elements in the coating will lead to the degradation of the coating (e.g., increasing the coating's porosity or be failed to form a densely top oxide layer), thus losing its protective effect [17,38,51,52,72]. In comparison, the Cr diffusion into the Zr matrix or the reaction diffusion between the Cr coating and the Zr substrate is not as serious as the case for some other coatings mentioned above. Although the interdiffusion effect can be mitigated by adding a barrier layer (e.g., Mo [64], TiC [73], and CrN [45] interlayers are known to work) to form a multilayer architecture, it is not conducive to the economical preparation and quality control of the coating.

From the perspective of practical application, because the coated cladding is always under normal operating conditions unless an accident happens, the interdiffusion between the coating and the substrate is very slow at relatively lower temperature, which is not expected to impose an obviously adverse impact on the overall performance of the cladding. Under accident conditions, although the interdiffusion process is accelerated, the coating still plays a role in delaying the progress of serious accidents to a certain extent due to its high resistance to high-temperature steam oxidation. However, it must be admitted that it is only an expedient measure to improve the accident resistance of fuel elements by coating the surface of zirconium alloy claddings, and more efforts are needed to develop new materials that can completely replace zirconium alloys.

## 5. Summary and Prospect

The Cr coatings on the Zr alloy are the most perspective material of choice in the ATF coating field. These materials belong to mainstream research worldwide and it is expected to greatly improve the safety of the reactor and fuel. The central focus of the research of the coating technology of Zr alloys has shifted from the stage of screening the coating composition several years ago to the stage of engineering development of metallic Cr coatings, whereas many engineering and scientific issues remain.

In general, the ATF cladding is a recently proposed emerging concept. The application of coatings to nuclear fuel claddings is in its development in many countries and more systematic and in-depth research is needed to define standardized preparation technology and evaluation procedures. Besides, the coating may solve many problems that cannot be solved in a single material, such as the hydrothermal dissolution of the $SiC_f$/SiC composite, the tritium penetration of FeCrAl, and severe volatilization of Mo oxides, which are mentioned in part I of the review. Therefore, the application of coating technology in nuclear reactor is the current development trend, which is worthy of further research efforts.

**Author Contributions:** Investigation, H.C.; Data curation, H.C.; Formal analysis, H.C.; Visualization, H.C.; writing—original draft preparation, H.C.; writing—review and editing, X.W. and R.Z.; Supervision, X.W. and R.Z.; Project administration, R.Z.; Funding acquisition, R.Z. All authors have read and agreed to the published version of the manuscript.

**Funding:** This research was funded by State Administration of Science, Technology and Industry for National Defense, PRC, Grant number 20181722.

**Acknowledgments:** The review presented was supported by the Research on Key Technology of Accident Tolerant Fuel of the Nuclear Energy Development Project in the State Administration of Science, Technology and Industry for National Defense, PRC. Shaoyu Qiu, Peinan Du, Weitian Guo, Yu Wang, and Hongyan Yang, from the Science

and Technology on Reactor Fuel and Materials Laboratory, Nuclear Power Institute of China, are acknowledged for providing aids in the investigation, revision and publication of this paper. Jean-Christophe Brachet (CEA, Univ. Paris-Saclay), Jeremy Bischoff (Framatome), Martin Ševeček ( Czech Technical University in Prague), John Fabian (American Nuclear Society) and Kirsten Epskamp (European Nuclear Society) are also acknowledged for providing suggestions in the copyrights permission.

**Conflicts of Interest:** The authors declare no conflict of interest.

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
