# Peer review of "Application and Development Progress of Cr-Based Surface Coating in Nuclear Fuel Elements: II. Current Status and Shortcomings of Performance Studies"

_coatings, doi:10.3390/coatings10090835_

Round 1
Reviewer 1 Report
- Comparisons with recent radiation resistance of Cr2AlC coatings should be made in accordance with most recent literature on the topic.
- Can the authors comment on the toxicity of Cr to human health and the impact of using Cr coatings in nuclear from the point of view of environment?
- Regarding Cr coating under accident conditions:
- Does pure Cr generates H gas upon oxidation with steam?
- Can the authors comment on the eutetic reaction between Cr-Zr at temperatures beyond 1300oC?
- Do we want coating zirconium alloys with materials that can react with the substrate at higher temperatures? E.g. formation of ZrCr2 intermetallic?
- The authors seem to be pro-Cr coating, but significantly neglect several other alternatives found in recent literature.
Reviewer 2 Report
Report to Coatings.
Application and Development Progress of Surface Coating in Nuclear Fuel Elements: II. Current Status and Shortcomings of Performance Studies on Cr-Based Coating
Zr alloys with optimized corrosion resistance, suitable mechanical properties, and a low thermal neutron absorption cross-section have been successfully used as a fuel cladding material in light water reactors (LWRs) for over four decades [1]. However, since the Fukushima-Daiichi accident in 2011, Zr alloys have suffered a huge challenge because of the poor oxidation resistance and the associated hydrogen gas generation in the loss-of-coolant accident or beyond-design-basis accident scenarios. Recently, the development of accident tolerant fuel is a critical issue in light water reactors. Cr coatings display a promising application in protecting Zr alloys from oxidation and maintaining fuel performance. The article review focused on accident tolerant fuel (ATF), which is an interesting topic mostly after the Fukushima nuclear accident in Japan.
Overall, I believe that this manuscript is a valuable contribution to the literature. I thus recommend publication, but only after my comments are addressed.
Comment
Since the fuel cladding with Uranium burn-up to 40-50 MW day/kg will be exposed to a damaging dose of ~10 dpa within 2 years and a maximum dose of ~25 dpa in 5 years.
In paragraph 2.4. Irradiation performance, the authors should add more details about irradiation with respect to these references:
- Microstructure evolution under light ions irradiation Hydrogen and He: Cr coating resist against damage and keep its crystallinity and He-H make a synergy to form bubbles (Scripta Materialia 187 (2020) 291–295); Journal of Nuclear Materials 538 (2020) 152240), on the other hand, they cause amorphization in SiC (Acta Materialia 188 (2020) 609-622; Acta Materialia 181 (2019) 160- 172).
- The corrosion behavior of irradiated Cr coatings (Journal of Alloys and Compounds 784 (2019) 1221-1233)
Could the authors make the caption of x-axis large in Fig.5 and Fig.12
With My best regards
Reviewer 3 Report
A very good, concise review on the issues of manufacturing, properties, service life and degradation mechanisms of Cr-based protective layers on zirconium alloys. Based on the review, the authors additionally formulate new concepts for the further development of coatings for the nuclear industry.
line 851: instead of Zs it should be Zr
line 1021: Czech Republic should be instead of Czech republic
line 1023: useless space
Author Response
- line 851: instead of Zs it should be Zr
revised as requested
- line 1021: Czech Republic should be instead of Czech republic
revised as requested
- line 1023: useless space
revised as requested
Round 2
Reviewer 1 Report
The revised paper is ok for publishing.